# Health outcome priorities of people with multiple long-term conditions using the outcome prioritisation tool in the UK: A survey study and feasibility assessment

Harini Sathanapally[1]*, Yogini V. Chudasama[2], Francesco Zaccardi[2], Alessandro Rizzi[2], Samuel Seidu[2], Kamlesh Khunti[2]

1 Diabetes Research Centre, Leicester General Hospital, University of Leicester, Leicester, United Kingdom, 2 Leicester Real World Evidence Unit, Diabetes Research Centre, Leicester General Hospital, University of Leicester, Leicester, United Kingdom

* hw326@leicester.ac.uk

## Abstract

### Background

The outcome prioritisation tool (OPT) is a simple tool to ascertain the health outcome priorities of people with MLTC. Use of this tool in people aged under 65 years with MLTC has not previously been investigated. This study aimed to investigate the feasibility of using the OPT in people with MLTC aged 45 years or above, in a multi-ethnic primary-care setting and describe the health outcome priorities of people with MLTC by age, clusters of long-term conditions and demographic factors, and to investigate any differences in prioritisation in light of the COVID-19 pandemic.

### Methods

This was a multi-centre cross-sectional study using a questionnaire for online self-completion by people aged 45 years or above with MLTC in 19 primary care settings across the East Midlands, UK. Participants were asked to complete the OPT twice, first from their current perspective and second from their recollection of their priorities prior to COVID-19.

### Results

The questionnaire was completed by 2,454 people with MLTC. The majority of participants agreed or strongly agreed that the OPT was easy to complete, relevant to their healthcare and will be useful in communicating priorities to their doctor. Summary scores for the whole cohort of participants showed *Keeping Alive* and *Maintaining Independence* receiving the highest scores. Statistically significant differences in prioritisation by age, clusters of long-term conditions and employment status were observed, with respondents aged over 65 most likely to prioritise *Maintaining independence*, and respondents aged under 65 most likely to prioritise *Keeping alive*. There were no differences before or after COVID-19, or by ethnicity.

**Data Availability Statement:** Relevant summarised data are in the manuscript or Supporting information files. Raw data cannot be

shared publicly as per data handling restrictions in view of potentially sensitive patient data being collected, set out as part of the ethical approval process. Ethical approval was received from Riverside REC Committee (Reference:20/LO/0570). Data requests may be sent to Riverside REC Committee and their contact email is riverside. rec@hra.nhs.uk.

**Funding:** The author(s) received no specific funding for this work.

**Competing interests:** KK is supported by the National Institute for Health Research (NIHR) Applied Research Collaboration East Midlands (ARC EM), NIHR Global Research Centre for Multiple Long-Term Conditions, MLTC Cross NIHR Collaboration (CNC) and the NIHR Leicester Biomedical Research Centre (BRC). SS, YC, FZ and HS are supported by NIHR ARC EM. This does not alter our adherence to PLOS ONE policies on sharing data and materials.

**Abbreviations:** CI, Confidence Interval; CRF, Case Report Form; CT, Clinical Trials; EC, Ethics Committee (see REC); GP, General Practice; ICF, Informed Consent Form; MLTC, Multiple long-term conditions; NHS, National Health Service; NRES, National Research Ethics Service; OPT, Outcome prioritisation tool; PPI, Patient and Public Involvement; R&D, NHS Trust R&D Department; REC, Research Ethics Committee; SD, Standard Deviation; UK, United Kingdom.

## Conclusions

The OPT is feasible and acceptable for use to elicit the health outcome priorities of people with MLTC across both middle-aged and older age groups and in a UK setting. Individual factors could influence the priorities of people with MLTC and must be considered by clinicians during consultations.

## Introduction

Finding ways of supporting people in managing multiple long-term conditions (MLTC), also known as multimorbidity [1], is now a global priority for policy and healthcare research [2]. The delivery of person-centred care, with the incorporation of patients' priorities and preferences into decision-making, is key to supporting people with effective management of MLTC's [3]. The National Institute of Health and Care Excellence (NICE) highlights the inclusion of patient's priorities, values and goals, in their quality standards for the management of multimorbidity [4]. Our previous systematic review showed there were major differences in the processes of prioritisation of people with MLTC and from the clinicians who were managing people with MLTC, thus highlighting the need to elicit the priorities of people with MLTC in each consultation, to avoid the risk of a misalignment of priorities [5]. Our review also found large variations in how priorities were ascertained, along with the tools used for this purpose, and we emphasised the need for a standardised tool to facilitate clinicians in eliciting the priorities of patients with MLTC [5].

The outcome prioritisation tool (OPT) previously developed by Fried et al, emerged as the most commonly used and validated tool to date in our systematic review [5, 6]. The OPT is a simple tool to facilitate clinicians in ascertaining the health outcome priorities of their patients across four domains, namely: *maintaining independence*, *keeping alive*, *reducing pain*, or *reducing other symptoms* [6]. Psychometric property testing was previously carried out by Fried et al, who found the tool was understood well by participants, with demonstrable construct validity and variable results in test re-test validity testing [6, 7]. The feasibility of using the OPT to elicit the priorities of people with MLTC aged 65 years and over [6], to facilitate medication review [8, 9], and complex treatment decision-making in older people with MLTC [10], has previously been investigated in clinical settings in the United States and in the Netherlands. However, this tool is yet to be applied to a population in the United Kingdom (UK), to younger people aged under 65 years and to an ethnically diverse population. Whilst a higher proportion of older adults suffer from MLTC [11], the prevalence of MLTC in younger adults is also high and continuing to rise [12]. Ethnicity-based differences in the risk of diagnoses of MLTC have also been found, with people of Asian ethnicity at increased risk of living with MLTC's [13]., The COVID-19 pandemic has raised further challenges for people with MLTC, as the presence of MLTC is an independent risk factor for an increased risk of severe COVID-19 and mortality [14, 15]. We therefore hypothesise that the COVID-19 pandemic may have had a further impact on the health outcome priorities of people with MLTC, and the domain of *keeping alive* may have been more highly prioritised by people with MLTC.

Therefore, this study aimed to describe the health outcome priorities of multi-ethnic people with MLTC including those aged 65 years and under using the OPT, investigate the association between patient's current first-choice health priority and risk factors, compare the health outcome priorities of patients before the onset of the COVID-19 outbreak to their current health

outcome priorities. We also aimed to examine the feasibility of the validated OPT by patient-reported relevance, ease of use, and patient-perceived usefulness in a UK population aged 45 or older.

## Methods

### Study design

This was a multi-centre, cross-sectional study using a questionnaire for self-completion by people aged 45 years and above, and living with at least two long-term conditions across the East Midlands, UK. Data was collected from August 2020 to January 2022. NHS Ethical approval for this study was granted by the Riverside REC Committee (Reference:20/LO/0570). The survey required about 10 minutes for respondents to fill out and assessed participants' demographic information, encompassing age, sex, ethnicity, self-reported long-term medical conditions, the quantity of regularly prescribed medications, and self-reported status of NHS identification as being at very high risk or extremely vulnerable from COVID-19. The questionnaire was initially to be completed by participants in a paper format after face-to-face recruitment; however, due to the COVID-19 pandemic, the questionnaire was adapted to an online format.

Potential participants were identified by members of the direct care team in the participating primary care practices using an electronic search of practice records for people fulfilling the inclusion criteria: aged 45 or above and had two or more coded diagnoses of long-term conditions, including cardiovascular, respiratory, metabolic, mental health, neurological, musculoskeletal and chronic pain conditions. Eligible participants were sent a brief invitation message introducing the study, a link to the participant information sheet and the online questionnaire via text or e-mail by the administration teams. The responses were captured and stored in a secure database through the REDCap software [16]. We did not collect any identifiable data from participants and therefore the completed questionnaires were fully anonymised from the outset. A formal sample size estimate was not set as it was decided a priori that sample size would be driven by response rate. We carried out a retrospective review of power for any statistical comparisons made within the data set.

### Health outcome prioritisation tool

Participants' health outcome priorities were measured using the OPT [6], namely: *maintaining independence*, *keeping alive*, *reducing pain*, and *reducing other symptoms*. Participants were asked to prioritise and rank the four health outcomes using a scale of 0–100, where 0 is the least priority and 100 is the high priority. They were asked to complete the tool twice, first considering their current priorities and second considering their priorities before the COVID-19 pandemic. The health outcome priorities have previously shown outstanding reliability on prior testing but the VAS scores of 0–100 demonstrated substantial variation, therefore for the analysis we chose a priori to analyse rank orderings only, i.e., first-choice only [17]. Due to the principle of trade-offs, if participants ranked two outcomes as the same i.e., 100, 100, or had any missing outcomes, these data were excluded from the analysis for the rankings to adhere to the OPT goals [18], hence this is the reduced cohort of the study population. After completing the OPT, participants were then asked to complete three Likert scale questions (ranging from strongly agree to strongly disagree) to measure the relevance, ease of use and patient-perceived usefulness of the OPT.

### Patient and public involvement engagement

We conducted a patient and public involvement (PPI) focus group with participants with MLTC aged 45 years or above and from a diverse range of ethnic backgrounds, to review the study documentation, including the participant information sheet and questionnaire. The purpose of this session was to gather input on the clarity and comprehensibility of these documents, as well as to gather any recommendations for enhancements. The feedback obtained from this session was used to amend the questionnaire to improve its readability and ease of understanding.

### Statistical analysis

The characteristics of the respondents and self-reported long-term conditions were presented as number (%), mean [standard deviation (SD)], or median [interquartile range (IQR)]. We described the health outcome priority scores in the whole cohort using the median [IQR]. In those who applied the trade-off principle (reduced cohort), we then examined the proportions of the rankings given as first-choice from the four health outcomes with further breakdown by the sociodemographic factors and clusters of conditions and the differences were tested using chi-squared statistics. The association between the respondent's first-choice health priority and sociodemographic factors and clusters of long-term conditions was estimated using logistic regression models. The odds ratio (OR) and 95% confidence intervals (CI) were reported comparing each factor and cluster of condition by each of the four health priority outcomes as first-choice, where an OR <1 indicated a lower health priority, whereas OR >1 indicated a higher health priority. For our main analysis, we report the adjusted models for age (continuous) and gender (female, male), as this allowed us to maintain a more robust and interpretable analysis. We also reported the unadjusted and the fully adjusted models using the categorical variables. We used the Hosmer-Lemeshow test to evaluate the model fit. To assess whether the health outcomes differed before COVID-19 and the current choice, we compared the continuous before and after scores for each outcome using the Wilcoxon signed ranked test. The feasibility analysis was carried out in the whole cohort which tested peoples' 'perceived usefulness' of the OPT. The three Likert scale responses ranged from strongly agree, agree, neither agree nor disagree, disagree, or strongly disagree, this was compared using chi-squared statistics. All statistical analyses were performed in Stata/IC 17.0; results were reported with 95% CI and statistical significance was defined at P<0.05. The statistical code is available at GitHub *yc244*.

## Results

### Survey respondents

Overall, 2,454 people with MLTC completed the survey from 19 primary care practices across the East Midlands, UK, S1 Fig in S1 File. The mean [SD] age was 63.6 [10.1] years, 57.1% were aged 45–65 years, and 42% were aged over 65 years; 58% were female, 92% were of White ethnicity, 36% were working, 8% were unemployed and 47% were retired, Table 1. A further breakdown of these categories is provided in S1 Table in S1 File. The median [IQR] number of self-reported long-term conditions was 3 [IQR 2–4], with the most common conditions being hypertension (48%), arthritis (36%), chronic pain (29%), depression (27%), anxiety (27%), and diabetes (26%), S2 Table in S1 File. When examining the clusters of long-term conditions, the largest was for cardiometabolic conditions (65%), musculoskeletal or chronic pain (52%), mental health (36%), and respiratory diseases (33%), Table 1. Regular medications were taken by 94% of the cohort, with the median [IQR] number of medications being 5 [3–7]. Almost 70% were at high risk of COVID-19, Table 1.

**Table 1. Characteristics of all survey respondents.**

| Characteristics | Total | Applied trade-off principle to OPT | | P-value |
|---|---|---|---|---|
| | | No | Yes | |
| **Total** | 2,454 | 1,570 (64.0) | 884 (36.0) | |
| **Age, y, mean [SD]** | 63.6 [10.1] | 63.4 [10.1] | 64.0 [10.0] | 0.095 |
| 45–64 y | 1,400 (57.1) | 917 (58.4) | 483 (54.6) | |
| ≥ 65 y | 1,036 (42.2) | 638 (40.6) | 398 (45.0) | |
| Missing | 18 (0.7) | 15 (1.0) | 3 (0.3) | 0.032 |
| **Gender** | | | | |
| Female | 1,412 (57.5) | 921 (58.7) | 491 (55.5) | |
| Male | 997 (40.6) | 618 (39.4) | 379 (42.9) | |
| Prefer not to say or missing | 45 (1.8) | 31 (2.0) | 14 (1.6) | 0.208 |
| **Ethnicity** | | | | |
| White | 2,251 (91.7) | 1,430 (91.1) | 821 (92.9) | |
| Non-white | 137 (5.6) | 90 (5.7) | 47 (5.3) | |
| Missing | 66 (2.7) | 50 (3.2) | 16 (1.8) | 0.114 |
| **Education** | | | | |
| None | 445 (18.1) | 304 (19.4) | 141 (16.0) | |
| GCSE, A Levels or equivalent | 1,165 (47.5) | 741 (47.2) | 424 (48.0) | |
| Higher education | 615 (25.1) | 376 (24.0) | 239 (27.0) | |
| Other (e.g., NVQ, nursing, missing) | 229 (9.3) | 149 (9.5) | 80 (9.1) | 0.113 |
| **Employment status** | | | | |
| Working | 887 (36.2) | 590 (37.6) | 297 (33.6) | |
| Unemployed | 196 (8.0) | 127 (8.1) | 69 (7.8) | |
| Retired | 1,143 (46.6) | 698 (44.5) | 445 (50.3) | |
| Other (student, volunteer, missing) | 228 (9.3) | 155 (9.9) | 73 (8.3) | 0.040 |
| **Taking regular medication** | | | | |
| Yes | 2,301 (93.8) | 1,465 (93.3) | 836 (94.6) | |
| No | 136 (5.5) | 92 (5.9) | 44 (5.0) | |
| Missing | 17 (0.7) | 13 (0.8) | 4 (0.5) | 0.360 |
| **Number of medications, median [IQR]** | 5 [3–7] | 5 [3–7] | 5 [3–7] | 0.423 |
| Polypharmacy (≥5 medications) | 1,318 (58.8) | 846 (59.5) | 472 (57.7) | 0.199 |
| **At very high risk of COVID-19** | | | | |
| Yes | 1,667 (67.9) | 1,062 (67.6) | 605 (68.4) | |
| No | 757 (30.9) | 487 (31.0) | 270 (30.5) | |
| Missing | 30 (1.2) | 21 (1.3) | 9 (1.0) | 0.753 |
| **No. conditions\*, median [IQR]** | 3 [2–4] | 3 [2–4] | 3 [2–4] | 0.148 |
| Cardiometabolic | 1,603 (65.3) | 1,019 (64.9) | 584 (66.1) | 0.563 |
| Musculoskeletal or chronic pain | 1,281 (52.2) | 801 (51.0) | 480 (54.3) | 0.118 |
| Mental health | 883 (36.0) | 559 (35.6) | 324 (36.7) | 0.604 |
| Respiratory | 819 (33.4) | 528 (33.6) | 291 (32.9) | 0.720 |

Shown are the number of subjects (%) unless stated otherwise. P-value indicates the difference between groups (Chi-square test); and continuous data (Wilcoxon rank sum test).

Y = years; SD = standard deviation; No. = number; IQR = interquartile range.

\* Self-reported long-term conditions. Cardiometabolic: diabetes, heart disease, atrial fibrillation, high blood pressure, stroke.

Musculoskeletal and chronic pain: arthritis, osteoporosis, long-term pain.

Mental health: depression and anxiety.

Respiratory: chronic obstructive pulmonary disease, asthma, and obstructive sleep apnoea.

Overall, 64% (n = 1,570) of respondents were not able to apply the trade-off principle to all the health outcomes as they either assigned the same score to more than one outcome or had a missing outcome. 36% (n = 884) of participants were able to apply the trade-off principle and gave a hierarchy of scores to all the health outcomes, S1 Fig in S1 File. The characteristics of those who did and did not apply the trade-off principle were similar, apart from the stratified age and employment groups, Table 1.

### Summary of health outcome priority scores

In the whole cohort (N = 2,454), the health outcome of *keeping alive* was the highest priority with the median value of 98 [IQR 88–100], followed by *maintaining independence* (97 [85–100]), *reducing pain* (92 [77–99]), and *reducing other symptoms* (87 [74–99]), Fig 1.

### Analysis of first-choice health outcome priority

**Current first-choice health outcome priority.** In the cohort of participants who did apply the trade-off principle, (n = 884), *keeping alive* was the outcome top-ranked as first-choice (44%), next was *maintaining independence* (33%), then *reducing pain* (14%), and *reducing other symptoms* (10%), S3 Table in S1 File. When comparing by the respondent's sociodemographic factors and clusters of conditions, we found statistically significant differences between most groups, S3 Table in S1 File. This was further illustrated by the significant

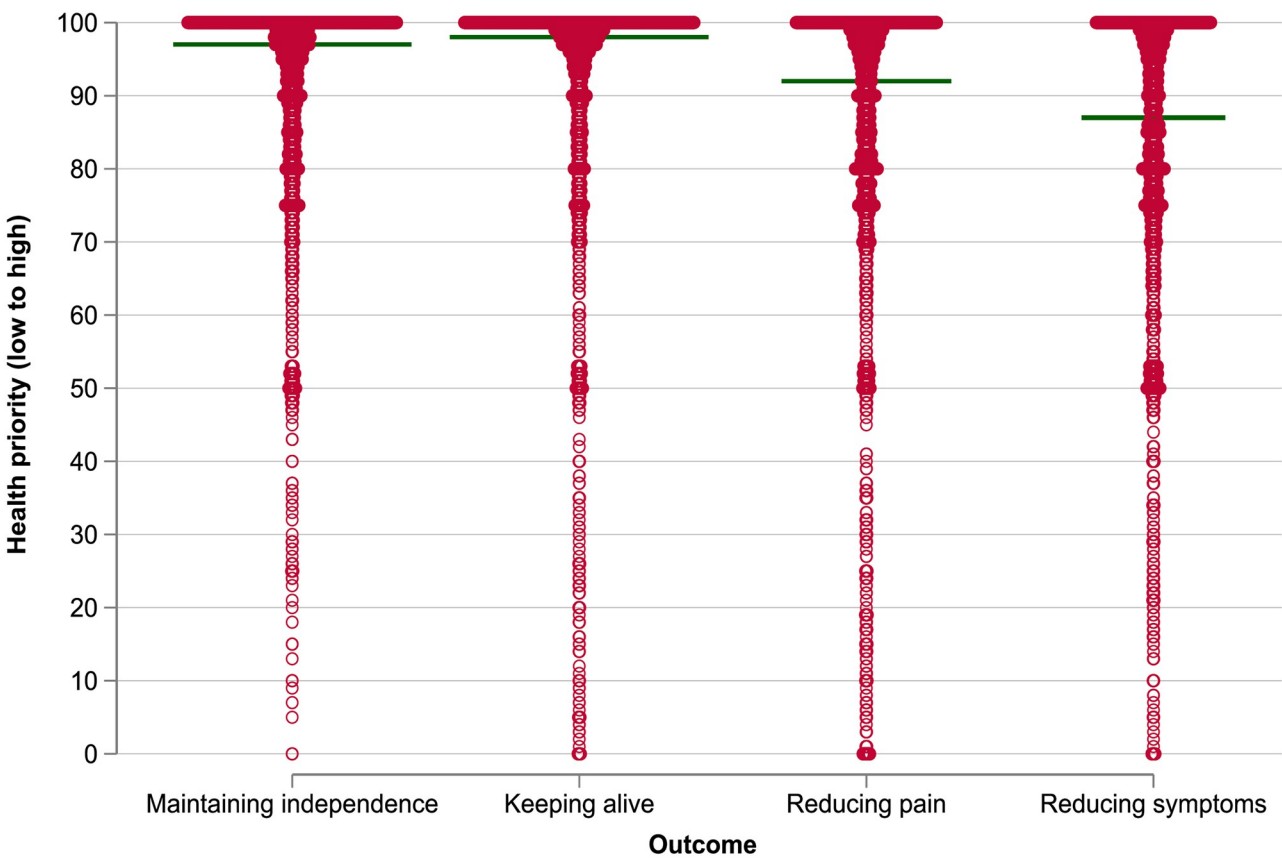

**Fig 1. Health outcome priority score in the whole cohort.** Reference line represents the median value of health prioritisation on the scale of 0 to 100. Median [interquartile range]: maintaining independence 97 [85–100], keeping alive 98 [88–100], reducing pain 92 [77–99], reducing other symptoms 87 [74–99]. Missing outcome: maintaining independence 140, keeping alive 164, reducing pain 221, reducing other symptoms 269.

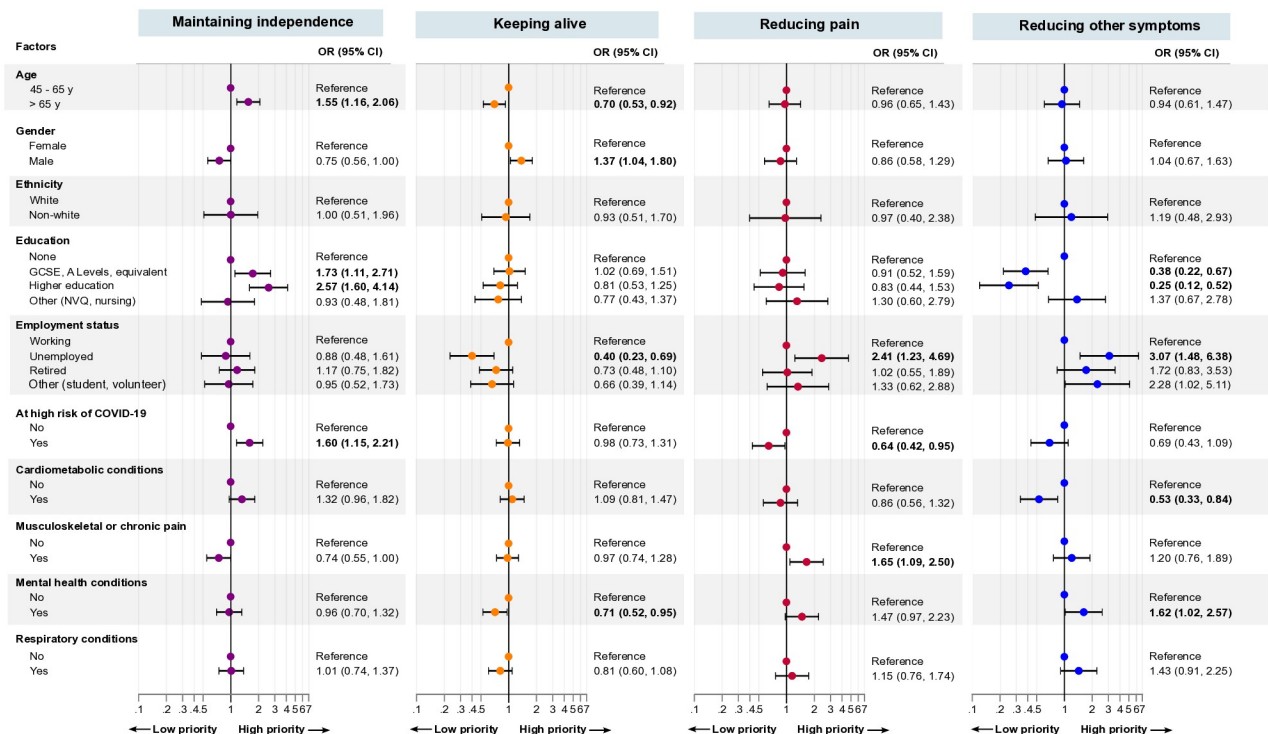

**Fig 2. Association between patient's first-choice health priority and sociodemographic factors and clusters of long-term conditions.** Models are adjusted by age (continuous) and gender (female or male). CI = confidence interval. Odds ratio <1 indicates low health priority, whereas odds ratio >1 indicate high health priority. Bold indicates statistical significance, P<0.05.

associations (S4, S5 Tables in S1 File, Fig 2). The results from our main analyses, adjusting for age and gender, we found that *maintaining independence* was highly prioritised for those aged 65 years or more (adjusted OR 1.55 (95% CI 1.16, 2.06)) compared to aged 45–65 years; with higher education (2.27 (1.60, 4.14)), or secondary/further education (1.73 (1.11, 2.71)), compared with no education; and those at high risk of COVID-19 (1.60 (1.15, 2.21)). Male respondents were 37% more likely to prioritise *keeping alive* compared to females (1.37 (1.04, 1.80)).

Respondents who were unemployed prioritised *reducing other symptoms* three times more than those working (3.07 (1.48, 6.38)), and *reducing pain* two times more (2.41 (1.23, 4.69)), and least prioritised *keeping alive* (0.40 (0.23, 0.69)). For those with musculoskeletal or chronic pain, their first-choice priority was *reducing pain* (1.65 (1.09, 2.50), and their least priority was *maintaining independence* (0.74, (0.55, 1.00)). While respondents who reported mental health conditions prioritised *reducing other symptoms* (1.62 (1.02, 2.57)), and least prioritised *keeping alive* (0.71 (0.52, 0.95)). Comparison by respondents' ethnicity did not reveal any statistically significant differences in prioritisation, S5 Table in S1 File, Fig 2. The result estimates were in a similar direction in the fully adjusted models, S6 Table in S1 File.

**Comparing the first-choice health outcome priorities before COVID-19 to current.** From a "before COVID-19" perspective the outcome most frequently ranked as the first-choice was *keeping alive* (38%), next *maintaining independence* (34%), then *reducing pain* (18%), and *reducing other symptoms* (10%), S7 Table in S1 File. There were no significant differences in health outcome priority before COVID-19 and current using the continuous scores, further evidenced in S2 and S3 Figs in S1 File.

**Table 2. Feasibility questions of patient's perceived usefulness of the health outcome prioritisation tool.**

| Feasibility question | Total | Applied trade-off principle to HOPT | | P-value |
|---|---|---|---|---|
| | **N = 2,454** | **No** | **Yes** | |
| | | **N = 1,570** | **N = 884** | |
| **1) The health outcome prioritisation tool was easy to complete** | | | | |
| Strongly agree | 563 (22.9) | 400 (25.5) | 163 (18.4) | |
| Agree | 1,108 (45.2) | 697 (44.4) | 411 (46.5) | |
| Neither agree nor disagree | 489 (19.9) | 305 (19.4) | 184 (20.8) | |
| Disagree | 147 (6.0) | 71 (4.5) | 76 (8.6) | |
| Strongly disagree | 133 (5.4) | 83 (5.3) | 50 (5.7) | |
| Missing | 14 (0.6) | 14 (0.9) | 0 (0.0) | <0.001 |
| **2) The health outcome prioritisation was relevant to my healthcare** | | | | |
| Strongly agree | 308 (12.6) | 232 (14.8) | 76 (8.6) | |
| Agree | 1,089 (44.4) | 675 (43.0) | 414 (46.9) | |
| Neither agree nor disagree | 772 (31.5) | 487 (31.0) | 285 (32.3) | |
| Disagree | 183 (7.5) | 111 (7.1) | 72 (8.2) | |
| Strongly disagree | 88 (3.6) | 53 (3.4) | 35 (4.0) | |
| Missing | 14 (0.6) | 12 (0.8) | 2 (0.2) | <0.001 |
| **3) The health outcome prioritisation tool will be useful in communicating what my priorities are to my doctor** | | | | |
| Strongly agree | 378 (15.4) | 292 (18.6) | 86 (9.7) | |
| Agree | 1,092 (44.5) | 652 (41.5) | 440 (49.8) | |
| Neither agree nor disagree | 639 (26.0) | 401 (25.5) | 238 (26.9) | |
| Disagree | 217 (8.8) | 140 (8.9) | 77 (8.7) | |
| Strongly disagree | 118 (4.8) | 75 (4.8) | 43 (4.9) | |
| Missing | 10 (0.4) | 10 (0.6) | 0 (0.0) | <0.001 |

P-value indicates the difference between groups (Chi-square test).

## Feasibility of the health outcome prioritisation tool

In the whole cohort, the majority of respondents agreed or strongly agreed that the OPT was easy to complete (68%), was relevant to their healthcare (57%), and useful in communicating what their priorities are to their doctor (60%), Table 2. The feasibility results differed in those who did and did not apply the trade-off principle, Table 2. In those who applied the trade-off principle (n = 884), the tool was particularly favoured by respondents with primary or secondary school education, at high risk of COVID-19, and with cardiometabolic conditions, musculoskeletal conditions or chronic pain (S8-S10 Tables in S1 File).

## Discussion

Our study describes the health outcome priorities of a large cohort of participants and compares the health outcome priorities by age, clusters of long-term conditions, ethnicity and demographic factors. To our knowledge, this is the largest study to investigate the use of the OPT in people with MLTC, and the first study to investigate the use of the OPT in the UK, in a multi-ethnic population and people with MLTC aged under 65. This is also the first study to investigate whether the health outcome priorities of people with MLTC have changed in light of the COVID-19 pandemic.

With a mean participant age of 64 years, the cohort in our study can be considered to be representative of middle-aged people with MLTC [19]. The majority of participants agreed or

strongly agreed that the OPT was easy to complete, relevant to their healthcare and will be useful in communicating priorities to their doctor. Hence, the results show that the OPT is relevant, useful, and easy to complete for middle-aged people with MLTC.

Overall, we found *keeping alive* was most frequently ranked as the first-choice health outcome priority, with *maintaining independence* being second most likely to be ranked as the top priority. Previous studies using the OPT in people with MLTC with sample sizes ranging from 59 to 357, have found that *maintaining independence* was most frequently ranked as the top priority [6–9]. This difference from our overall findings could be since previous studies using the OPT have all focused on older people with MLTC, whereas we included middle-aged people with MLTC in our cohort. Indeed, a comparison of participants' ranking by age, revealed that participants aged over 65 were most likely to rank *maintaining Independence* as their top priority, and were more likely to do so than participants aged under 65 years, which is in line with previous literature [6, 7, 9].

[5] We also observed statistically significant differences in participants' priorities based on types of long-term conditions. Participants with musculoskeletal conditions or chronic pain were most likely to choose *reducing pain* as their first-choice health outcome priority. Previous literature has demonstrated that the empirical impact of illnesses including their symptom burden [5], and disease burden [20], were factors which influenced how people with MLTC chose priorities related to their health and could provide the likely explanation for this finding, as the symptom of pain has the potential to form a significant part of the symptom burden experienced by people with musculoskeletal conditions [21], and/or chronic pain. Participants with mental health conditions were most likely to choose *reducing other symptoms* as their top priority, and *keeping alive* the least, which suggests that for these participants, their health outcome priorities could be related to the symptom burden of their illness but these were not available as a choice to be captured on the OPT.

We also found that socio-economic factors such as education and employment status were associated with statistically significant differences in prioritisation. Adjusting for age and gender, our results indicated that participants who were unemployed were three times more likely to prioritise *reducing other symptoms*, and twice more likely to prioritise *reducing pain*, compared to those who were employed. Prolonged unemployment has previously been found to be associated with musculoskeletal pain [22]. A previous study in the context of multiple sclerosis found that the presence of pain was associated with a reduced rate of employment [23], and another study investigating the impact of pain on employment amongst cancer survivors found that pain was associated with an increased risk of adverse employment outcomes [24]. Our findings demonstrate that socio-economic factors can have an impact on the health outcome priorities of people with MLTC, and highlight the importance of taking a holistic person-centred approach that considers the impact of the socio-economic background of people with MLTC on an individual basis when eliciting their health outcome priorities in consultations.

We did not observe any statistically significant changes in participants' reporting of their health outcome prioritisation from before the COVID-19 pandemic and currently. However, these results are subject to recall bias, as participants were asked to retrospectively report their health outcome priorities from before the onset of COVID-19.

[6] We found that a significant number of participants (64%) gave the same scores to multiple domains or missed outcomes, and that three of the four health outcomes (*maintaining independence*, *keeping alive* and *reducing pain*) had received a median score of 92 or above. In these instances, as well as where participants assigned the same scores to multiple domains, it may be that participants placed an equal or almost equal personal value on multiple or all of those health outcomes. The fact that participants were being asked to consider their priorities

outside of a situation where there was an actual decision to be made rather than a hypothetical one, could also have been a contributing factor to this. A previous study investigating the use of the OPT for medication review for older people (aged 69 years or above) with MLTC found that participants reported engaging with prioritisation to be "difficult when there was no specific need to make a decision" [8]. Furthermore, whilst the participant information sheet explained that completing this questionnaire would have no impact at all on their healthcare, participants may also still have been concerned about any implications on their healthcare by choosing a particular domain as a higher priority over another.

## Strengths and limitations

To our knowledge, this was the first large study to implement the OPT in a UK setting, and the first to implement the OPT in participants from a multi-ethnic population aged under 65 years. This study has made a contribution to understanding the health outcome priorities of people with MLTC and the factors that may influence them, which is an important step towards facilitating the development of priorities-based models of care for people with MLTC. With participation from multiple primary care practices across the East Midlands, we found that the OPT acceptable to people with MLTC aged under 65 years, and can feasibly be used in future interventions to facilitate priorities-based care for people with MLTC aged 45 or above, in primary care settings.

However, there are noteworthy limitations. As this was a cross-sectional study conducted after the onset of the COVID-19 pandemic, participants' responses regarding their health outcome priorities prior to the COVID-19 pandemic were retrospective, and hence are subject to recall bias. As data collection continued until August 2022, the length of time to recall priorities from before the COVID-19 pandemic will have extended up to two and a half years for some participants who completed the questionnaire closer to the study closure date, which may have further potentiated the risk of recall bias.

The OPT has previously been used in a face-to-face format, where participants were helped by facilitators to indicate their order of prioritisation [6]. However, in this study, the OPT was disseminated in an online format in light of the COVID-19 pandemic, and whilst participants were advised in the participant information sheet to contact their practice or the study team if they had any questions regarding the study, direct facilitation to help patients complete the OPT was not feasible.

We also found that a significant number of participants assigned the same scores to multiple health outcomes, suggesting that the "trade-off" principle was not always understood by participants which could have been due to the virtual format of the questionnaire and the absence of a facilitator. The need to assign a score to each health outcome as part of the tool could also potentially lead to confusion regarding the trade-off principle.

Despite the study being set in a multi-ethnic setting with recruitment of primary care practices with high levels of ethnic diversity amongst their practice populations, we noted that there was a relatively low uptake of participation in our study from participants from ethnic minority backgrounds, with 92% of participants being of White ethnicity, and lower proportions of ethnic diversity amongst participants compared to National and Regional demographic data [25]. A possible explanation for this could be due to a language barrier, as the invitation to participate in the study, and the study questionnaire itself were not available in any other languages besides English at this stage.

The digital format of the study could also have introduced selection bias towards younger people with MLTC, as it required access to an electronic device to participate, there was a larger number of participants aged under 65 years (57%), compared to participants aged over

65 years (42%). Eligible patients with MLTC were identified by practices and sent the invitation link to participate in the study, however the diagnoses of long-term conditions, number of regular medications and status of NHS identification as being at very high risk or extremely vulnerable from COVID-19 were self-reported by participants and could have introduced potential inaccuracies in the data collected, which we cannot verify due to the data being anonymised from the point of collection. Lastly, our main analyses were adjusted for age and gender to maintain a more robust and interpretable analysis, however further research including a larger sample size would enable further confounders to be accounted for with a higher statistical power.

## Recommendations for the future

We suggest that the OPT could be sent remotely to patients with MLTC's ahead of their consultations, to allow them an opportunity to consider their health outcome priorities and feel prompted to discuss these during their consultations. During consultations, we suggest further exploration of patients' priorities using the patient's responses or reflections from the OPT as a starting point, and with consideration of the possibility that there may be multiple health outcomes that are prioritised highly or equally by patients, as well as health outcome priorities held by patients that have not been captured by the OPT. We also suggest that this process is revisited at any point where there is a management or treatment decision to be made. This approach would promote shared decision making and person-centred management during consultations with people with MLTC, and facilitate efficiency with allowing patients to consider their own health outcome priorities ahead of consultations.

To reduce the risk of confusion in assigning scores to different health outcomes and facilitate participants in applying the trade-off principle to reflect a hierarchy of health outcomes in order of importance to them on an individual basis, we propose a re-design of the OPT in which participants are asked to place each health outcome into four coloured boxes in order of their individual importance to them (Fig 3). We propose that this redesign would clearly capture the rank assigned to each health outcome by separating them into four distinct colours with the rank order assigned, as well as mitigate the potential for confusion through the removal of a need for participants to assign a score. We suggest further input and close co-production with representatives of patients living with MLTC's and healthcare providers including primary care clinicians, to develop strategies for the successful integration of this tool along with the proposed re-design into primary care consultations, with a focus on clear and effective communication with patients regarding the purpose and implications of the use of the tool for their healthcare in consultations.

We also recommend translating the OPT into different languages to address any potential language barrier for people with MLTC completing the OPT. We recommend further consultation and input from patient representatives and healthcare providers for co-development of strategies for successful integration of the OPT into consultations, and effective communication regarding the tool with patients. We suggest that the OPT could be used remotely ahead of consultations with people with MLTC to prompt and facilitate conversations about each patient's individual health outcome priorities during primary care consultations, as part of a shared decision-making process with person-centred management.

## Conclusion

We found that the validated OPT was acceptable to people aged 45 or over with MLTC and has the potential to be a means to introduce and incorporate individual priorities into their clinical consultations and facilitate person-centred, priorities-based care planning. We found

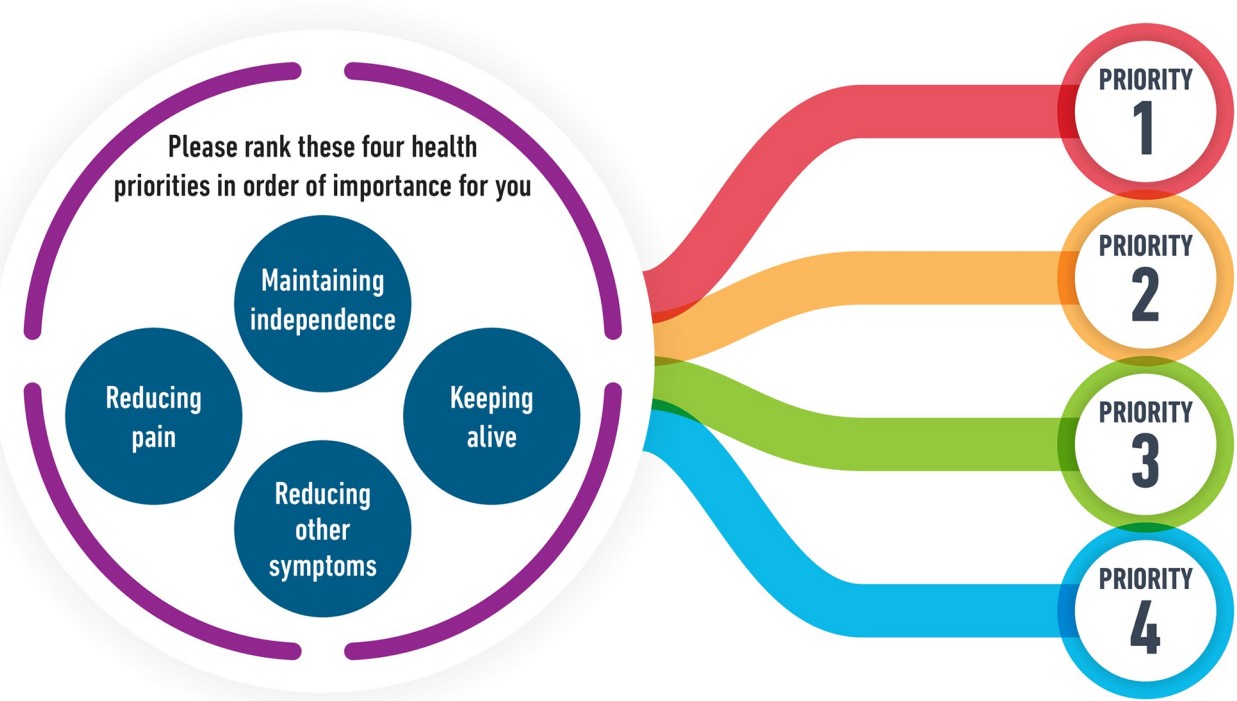

**Fig 3. Health outcome prioritisation tool simplified for clinical use.** Health outcome prioritisation tool developed by Fried et al [4].

that *keeping alive* was most frequently ranked as the first-choice health outcome priority across all our participants, with *maintaining independence* being second most likely to be ranked as the top priority. We found that the health outcome priorities of our participants were influenced by a number of factors individual to each patient such as age, types of long-term conditions, and employment status. We recommend that in their consultations with people with MLTC, clinicians should take a holistic approach that takes all of these individual factors into account.

## Supporting information

**S1 File.**
(DOCX)

## Acknowledgments

We would like to thank the East Midlands CRN for their support with this study.

## Author Contributions

**Conceptualization:** Harini Sathanapally, Samuel Seidu, Kamlesh Khunti.

**Data curation:** Harini Sathanapally.

**Formal analysis:** Harini Sathanapally, Yogini V. Chudasama, Francesco Zaccardi, Alessandro Rizzi.

**Investigation:** Harini Sathanapally.

**Methodology:** Harini Sathanapally.

**Project administration:** Harini Sathanapally.

**Software:** Yogini V. Chudasama.

**Supervision:** Francesco Zaccardi, Samuel Seidu, Kamlesh Khunti.

**Writing – original draft:** Harini Sathanapally, Yogini V. Chudasama.

**Writing – review & editing:** Harini Sathanapally, Yogini V. Chudasama, Francesco Zaccardi, Alessandro Rizzi, Samuel Seidu, Kamlesh Khunti.

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
