## [Decision Letter · Decision Letter 0]

27 May 2024

PONE-D-24-10844Health outcome priorities of people with multiple long-term conditions before and during the COVID-19 pandemic: Survey data from the UKPLOS ONE

Dear Dr. Sathanapally,

Thank you for submitting your manuscript to PLOS ONE. After careful consideration, we feel that it has merit but does not fully meet PLOS ONE’s publication criteria as it currently stands. Therefore, we invite you to submit a revised version of the manuscript that addresses the points raised during the review process.

We look forward to receiving your revised manuscript.

Kind regards,

Filipe Prazeres, MD, MSc, Ph.D.

Academic Editor

PLOS ONE

“KK is supported by the National Institute for Health Research (NIHR) Applied Research Collaboration East Midlands (ARC EM), NIHR Global Research Centre for Multiple Long-Term Conditions, MLTC Cross NIHR Collaboration (CNC) and the NIHR Leicester Biomedical Research Centre (BRC). SS, YC, FZ and HS are supported by NIHR ARC EM.”

Reviewers' comments:

Reviewer's Responses to Questions

**Comments to the Author**

1. Is the manuscript technically sound, and do the data support the conclusions?

Reviewer #1: Yes

Reviewer #2: Partly

Reviewer #3: Partly

Reviewer #4: Partly

2. Has the statistical analysis been performed appropriately and rigorously? 

Reviewer #1: Yes

Reviewer #2: Yes

Reviewer #3: Yes

Reviewer #4: Yes

3. Have the authors made all data underlying the findings in their manuscript fully available?

Reviewer #1: Yes

Reviewer #2: No

Reviewer #3: Yes

Reviewer #4: Yes

4. Is the manuscript presented in an intelligible fashion and written in standard English?

Reviewer #1: Yes

Reviewer #2: Yes

Reviewer #3: Yes

Reviewer #4: Yes

5. Review Comments to the Author

Reviewer #1: Dear Author

Thank you very much for the peer review opportunity.

I read your paper with great interest.

I think this study is very useful for the future care of MLTC patients.

1

What do you think about statistically testing Tables 1 and 2 ? I think it would help the reader to better understand the results.

2

I believe one of the object of this study was to examine changes in health outcome priorities before and after the COVID-19 pandemic. What hypothesis would this be based on? If you present a hypothesis, the reader may be able to better understand the results.

3

It is suggested that when patients use the OPT in the future, they place each health outcome in four colored boxes in order of personal importance (Figure 3). I understand that the four color boxes were not used in this study. Did the patient offer any suggestions? The reader would better understand the results if you could provide the rationale for this proposal.

Reviewer #2: The submitted paper reports findings from a survey of health outcome priorities among people living with multiple long-term conditions. The analysis includes current priorities, and retrospectively recalled priorities prior to Covid-19. The perceived usefulness of the tool from the patient perspective is also reported. Comments for the authors to consider are included in chronological order, following a more general comment about health outcome priorities:

This paper carries an inherent assumption that people do or should prioritise health outcomes using a trade-off approach. While the value of communicating priorities is discussed in the introduction, this assumption needs to be explored further from a clinical and patient perspective. It is a finding in itself that more than half of participants did not apply this trade-off principle to their decision making, which would have implications for clinical management, e.g. people may have concerns that scoring keeping alive as a lower priority may have implications for their care. It is suggested by the authors that the survey implementation is redesigned to apply these choices more explicitly. Is it assumed that this would reflect a patient’s reality, though actually this study shows people are unable/unwilling to trade-off when not forced to. How should a clinician interpret ranked priorities if it’s entirely possible that a person values several outcomes equally? The review of decisions at the consultation phase is discussed, and this reflection could also be developed further there, and in the rationale for using this tool in practice. While it's welcome that the tool was rated as useful by users, I would suggest that a recommendation for future could be to work closely with public and patient engagement/involvement representatives and healthcare providers to ensure the framing and communication around this tool is careful and comprehendible when it is being presented to patients.

The title of the paper is slightly misleading given the retrospective nature of pre-covid assessment which is later stated as a key limitation and is only presented at the end of the results section. The aims state that the paper will describe health outcome priorities, compare them to pre-covid priorities, and determine the feasibility of the tool. Realigning the title to better reflect these broader aims is advised.

The introduction provides a rationale for the research, citing the novelty of focusing on the UK population and those under 65 years. The second paragraph repeats this idea a couple of times so could be edited for brevity.

Re language, I suggest rephrasing “management of people with multimorbidity” to “supporting people to manage” or “management of multimorbidity”. Here and later in the methods, I also suggest rephrasing “suffering from MLTCs” to “living with MLTCs”.

The data collection period is August 2020 to August 2022, meaning those at the later stage were recalling a longer time period to pre-covid. This could also be mentioned when considering this limitation in the discussion section.

The results are clearly presented in text and with tables and figures. It is stated that all relevant data are in the manuscript or supporting information files. It should be clarified that this is summarised data and not the full underlying dataset, with reason given if unable to publicly share the full dataset.

There is a ceiling effect with this tool which should be discussed more explicitly, with a median score of 92 or above for three of the outcomes. This highlights that none of the outcomes are necessarily of low priority to patients, which may be valuable information that is lost if the trade-off principle is applied.

Reviewer #3: 1 - Is the manuscript technically sound, and do the data support the conclusions?

Upon thorough examination of the manuscript, it can be stated that it presents scientifically sound research supported by robust data that underpins the drawn conclusions. The experiments conducted were rigorously carried out, incorporating appropriate controls, replications, and a suitable sample size, thereby enhancing the reliability of the obtained results. Furthermore, the conclusions presented in the study are appropriately derived from the analyzed data, demonstrating a careful and well-founded approach to interpreting the findings.

However, to enhance the transparency and completeness of the statistical analysis, it is suggested to provide more details regarding the selection of variables in the multivariate models and to discuss potential sources of bias. Additionally, a more detailed description of the health outcome prioritization tool used in the study, including its validity, reliability, and development, would be beneficial for readers to better evaluate the tool's utility and applicability in other clinical and research settings.

Furthermore, a dedicated section discussing the limitations of the study would be important to provide a critical view of the results, including discussions on potential selection biases, limitations of the methodology used, and issues related to the generalization of the findings. A discussion of the clinical and policy implications of the study results could also be included to highlight the practical relevance of the study, providing recommendations for clinical practice, public health policies, and directions for future research.

2 - Has the statistical analysis been performed appropriately and rigorously?

After reviewing the statistical analysis conducted in the manuscript, I noticed that the analysis was conducted appropriately, but there are areas where additional details and rigor could improve its comprehensiveness.

The text employs statistical techniques such as chi-square statistics, logistic regression models, and the Wilcoxon test to examine associations, differences, and changes in health priorities among participants. However, to provide a more comprehensive view of the statistical analysis, I have some suggestions for the authors:

Provide more details on the multivariate analysis, such as information on variable selection, consideration of interactions between variables, and interpretation of results in terms of the impact of different variables on health priorities.

Discussion of Potential Bias: A discussion of potential sources of bias in the statistical analysis, such as selection bias or confounding, would be important to assess the robustness of the results. Conducting sensitivity analyses to evaluate the potential impact of these biases on the results would further strengthen the analysis.

Model Evaluation: Providing information on how the adequacy of the statistical models was assessed, such as through model fit tests or cross-validation, would help ensure that the chosen statistical models are appropriate for the data and provide reliable results.

Control for Possible Confounding Factors: In addition to the variables adjusted in the logistic regression models, discussing whether other confounding factors were considered in the statistical analysis and how these factors were controlled for would ensure more accurate and reliable results.

______

Based on this analysis, the manuscript appears to be technically sound, with the data presented supporting the conclusions drawn. The study seems to have been conducted rigorously, with adequate controls and appropriate statistical analysis. However, it would be useful to have more details on the statistical methodology used, especially regarding the multivariate analysis of associations between health priorities and risk factors.

Reviewer #4: Congratulations to the authors!

This is a good-quality manuscript. The research question is important. However, I will make some comments and ask some questions as a contribution to the authors in order to improve the work.

- Overall, the authors presented a good justification for the reported investigation. The main objective is not entirely clear, as it diverges slightly from the main idea set out in the summary. In the methods description, they presented the study location, including periods and methods of collecting data from the participants. The discussion could be a little more in-depth. The results provided a descriptive analysis of the study participants, a good number of well-presented tables and figures. It presented some limitations and potential bias of the study. The conclusion did not emphasize the title and main objective of the study.

- The title does not refer to the evaluation of an OPT assessment instrument. However, throughout the work and especially in the conclusion, there was an emphasis on the OPT instrument, and not on the response to the main objective of the proposed study: to describe the health outcome priorities of people with MLTC by age, clusters of long-term conditions and demographic factors, and to investigate any differences in prioritization in light of the COVID-19 pandemic.

- The summary provided an informative synthesis of what was found; however, the title is not in line with the content presented in the summary that emphasizes the feasibility of using the OPT in people with MLTC aged 45 years or above. When reading the abstract, I thought this would be the main objective of the study.

- It is important to highlight that, regarding abbreviations, in the first citation it is necessary to put the name in full preceding it. This did not happen with the abbreviation “MLTC”.

Despite this, to describe the health outcome priorities of people with MLTC by age, clusters of long-term conditions and demographic factors, and to investigate any differences in prioritization in light of the COVID-19 pandemic, most of the Statistical methods were described properly. The authors presented characteristics of study participants. Main results adequately described.

- The methodology used in this study did not use criteria and statistical tests used to evaluate the reliability (stability, internal consistency and equivalence) and validity (content, criteria and construct) of instruments. Therefore, in this study design, it is not recommended to infer about the feasibility of using the OPT in people with MLTC aged 45 years or above.

Questions:

- How was the sample size estimated so that it was representative?

- Were there any exclusion criteria for participants?

- Would it be possible to discuss expected results before and after the pandemic and bring results from similar studies?

- Could you better explain the statement “they could be more aggressive in their clinical management, in people with MLTC aged under 65”?

- Wouldn't self-reported status of NHS be a limitation of the study and potentially cause bias?

- Would elderly patients have more difficulty answering online surveys? Please discuss if that would be a limitation of the study.

6. PLOS authors have the option to publish the peer review history of their article (what does this mean?). If published, this will include your full peer review and any attached files.

Reviewer #1: No

Reviewer #2: No

Reviewer #3: No

Reviewer #4: **Yes: **Januse Nogueira de Carvalho

---

## [Author Response · Author response to Decision Letter 0]

14 Oct 2024

Dear Author

Thank you very much for the peer review opportunity.

I read your paper with great interest.

I think this study is very useful for the future care of MLTC patients.

1. What do you think about statistically testing Tables 1 and 2? I think it would help the reader to better understand the results.

Response: Many thanks for your comments and time in reviewing our manuscript.

We agree that statistically testing the data in Tables 1 and 2 would benefit the reader, as this would better compare those who applied the trade-off principle to those who did not. We have now included the P-values in Table 1 and Table 2, and have updated the manuscript: 

Page 11 “The characteristics of those who did and did not apply the trade-off principle were similar, apart from the stratified age and employment groups, Table 1.”

Page 13 “The feasibility results differed in those who did and did not apply the trade-off principle, Table 2.”

2. I believe one of the objectives of this study was to examine changes in health outcome priorities before and after the COVID-19 pandemic. What hypothesis would this be based on? If you present a hypothesis, the reader may be able to better understand the results.

Response: Thank you, we have added the hypothesis to the manuscript:

Introduction – Page 7: "We therefore hypothesise that the COVID-19 pandemic may have had a further impact on health outcome priorities of people with multiple long-term conditions, and the domain of "keeping alive" may have been more highly prioritised by people with MLTC"

3. It is suggested that when patients use the OPT in the future, they place each health outcome in four colored boxes in order of personal importance (Figure 3). I understand that the four color boxes were not used in this study. Did the patient offer any suggestions? The reader would better understand the results if you could provide the rationale for this proposal.

Response: Thank you, we made this suggestion to reiterate the concept of assigning a hierarchy of prioritisation to the four health outcome domains by separating the hierarchy into four distinct colours. This suggestion has been made by the study team and not through any input from patients or members of the public. We have further clarified the rationale and added a suggestion for further co-development of this redesign of the tool in collaboration with advisory groups of people with MLTC in the manuscript:

Pages 20 and 21: "We propose that this redesign would clearly capture the rank assigned to each health outcome by separating them into four distinct colours with the rank order assigned, the participant as well as mitigate the potential for confusion through the removal of a need for participants to assign a score, and . we suggest further co-development of this redesign of the tool in collaboration with advisory groups of people with MLTC to facilitate effective integration into primary care clinical settings." 

REVIEWER #2

Comments Response 

The submitted paper reports findings from a survey of health outcome priorities among people living with multiple long-term conditions. The analysis includes current priorities, and retrospectively recalled priorities prior to Covid-19. The perceived usefulness of the tool from the patient perspective is also reported. Comments for the authors to consider are included in chronological order, following a more general comment about health outcome priorities:

This paper carries an inherent assumption that people do or should prioritise health outcomes using a trade-off approach. While the value of communicating priorities is discussed in the introduction, this assumption needs to be explored further from a clinical and patient perspective. It is a finding in itself that more than half of participants did not apply this trade-off principle to their decision making, which would have implications for clinical management, e.g. people may have concerns that scoring keeping alive as a lower priority may have implications for their care. It is suggested by the authors that the survey implementation is redesigned to apply these choices more explicitly. Is it assumed that this would reflect a patient’s reality, though actually this study shows people are unable/unwilling to trade-off when not forced to. How should a clinician interpret ranked priorities if it’s entirely possible that a person values several outcomes equally? The review of decisions at the consultation phase is discussed, and this reflection could also be developed further there, and in the rationale for using this tool in practice. While it's welcome that the tool was rated as useful by users, I would suggest that a recommendation for future could be to work closely with public and patient engagement/involvement representatives and healthcare providers to ensure the framing and communication around this tool is careful and comprehendible when it is being presented to patients.

 Response: We thank the reviewer for their valuable feedback. We have added further reflections on the reasons for why over half of participants had assigned the same scores to multiple domains, including the potential reasons raised by the reviewer on in the discussion section:

Discussion- Page 16 " We found that a significant number of participants (64%) gave the same scores to multiple domains or missed outcomes, and that three of the four health outcomes (maintaining independence, keeping alive and reducing pain) had received a median score of 92 or above. In these instances, as well as where participants assigned the same scores to multiple domains, it may be that participants placed an equal or almost equal personal value on multiple or all of those health outcomes. 

The fact that participants were being asked to consider their priorities outside of a situation where there was an actual decision to be made rather than a hypothetical one, could also have been a contributing factor to this. A previous study investigating the use of the OPT for medication review for older people (aged 69 years or above) with MLTC found that participants reported engaging with prioritisation to be “difficult when there was no specific need to make a decision” (9). Furthermore, whilst the participant information sheet explained that completing this questionnaire would have no impact at all on their healthcare, participants may also still have been concerned about any implications on their healthcare by choosing a particular domain as a higher priority over another."

We have also added a recommendation for further input and close co-production with patient representatives and healthcare providers in the recommendations for the future section of the manuscript and also restructured this section for better alignment with the rest of the manuscript:

Recommendtations for the future - Page 19: " We suggest that the OPT could be sent remotely to patients with MLTC’s ahead of their consultations, to allow them an opportunity to consider their health outcome priorities and feel prompted to discuss these during their consultations. During consultations, we suggest further exploration of patients' priorities using the patient's responses or reflections from the OPT as a starting point, and with consideration of the possibility that there may be multiple health outcomes that are prioritised highly or equally by patients, as well as health outcome priorities held by patients that have not been captured by the OPT. We also suggest that this process is revisited at any point where there is a management or treatment decision to be made. This approach would promote shared decision making and person-centred management during consultations with people with MLTC, and facilitate efficiency with allowing patients to consider their own health outcome priorities ahead of consultations. 

We suggest further input and close co-prodution with representatives of patients living with MLTC's and healthcare providers including primary care clinicians, to develop strategies for the successful integration of this tool along with the proposed re-design into primary care consultations, with a focus on clear and effective communication with patients regarding the purpose and implications of the use of the tool for their healthcare in consultations."

The title of the paper is slightly misleading given the retrospective nature of pre-covid assessment which is later stated as a key limitation and is only presented at the end of the results section. The aims state that the paper will describe health outcome priorities, compare them to pre-covid priorities, and determine the feasibility of the tool. Realigning the title to better reflect these broader aims is advised.

 Response: We have revised the title to:

"Health outcome priorities of people with multiple long-term conditions using the outcome prioritisation tool in the UK: a survey study and feasibility assessment"

The introduction provides a rationale for the research, citing the novelty of focusing on the UK population and those under 65 years. The second paragraph repeats this idea a couple of times so could be edited for brevity.

 Response: Thank you, we have edited the second paragraph accordingly 

Re language, I suggest rephrasing “management of people with multimorbidity” to “supporting people to manage” or “management of multimorbidity”. Here and later in the methods, I also suggest rephrasing “suffering from MLTCs” to “living with MLTCs”.

 Response: Thank you, we have made this amendment.

The data collection period is August 2020 to August 2022, meaning those at the later stage were recalling a longer time period to pre-covid. This could also be mentioned when considering this limitation in the discussion section.

 Response: Thank you, we accept this limitation and have added this point to the strengths and limitations section of the manuscript 

Strengths and Limitations- Page 17: " As this was a cross-sectional study conducted after the onset of the COVID-19 pandemic, participants’ responses regarding their health outcome priorities prior to the COVID-19 pandemic were retrospective, and hence are subject to recall bias. As data collection continued until August 2022, the length of time to recall priorities from before the COVID-19 pandemic will have extended up to two and a half years for some participants who completed the questionnaire closer to the study closure date, which may have further potentiated the risk of recall bias." 

The results are clearly presented in text and with tables and figures. It is stated that all relevant data are in the manuscript or supporting information files. It should be clarified that this is summarised data and not the full underlying dataset, with reason given if unable to publicly share the full dataset.

 Response: We have amended the data statement to "Relevant summarised data are in the manuscript or supporting information files. Raw data cannot be shared publicly as per data handling restrictions set out as part of the ethical approval process (ethical approval received from Riverside REC Committee (Reference:20/LO/0570))"

There is a ceiling effect with this tool which should be discussed more explicitly, with a median score of 92 or above for three of the outcomes. This highlights that none of the outcomes are necessarily of low priority to patients, which may be valuable information that is lost if the trade-off principle is applied.

 Response: We have added updated the manuscript with further reflections and recommendation regarding this point:

Discussion- Page 16: "We found that a significant number of participants (64%) gave the same scores to multiple domains or missed outcomes, and that three of the four health outcomes (maintaining independence, keeping alive and reducing pain) had received a median score of 92 or above. In these instances, as well as where participants assigned the same scores to multiple domains, it may be that participants placed an equal or almost equal personal value on multiple or all of those health outcomes."

Recommendations for the future- Page 19 : "During consultations, we suggest further exploration of patients' priorities using the patient's responses or reflections from the OPT as a starting point, and with consideration of the possibility that there may be multiple health outcomes that are prioritised highly or equally by patients, as well as health outcome priorities held by patients that have not been captured by the OPT."

REVIEWER #3

Comments Response 

1 - Is the manuscript technically sound, and do the data support the conclusions? 

Upon thorough examination of the manuscript, it can be stated that it presents scientifically sound research supported by robust data that underpins the drawn conclusions. The experiments conducted were rigorously carried out, incorporating appropriate controls, replications, and a suitable sample size, thereby enhancing the reliability of the obtained results. Furthermore, the conclusions presented in the study are appropriately derived from the analyzed data, demonstrating a careful and well-founded approach to interpreting the findings.

However, to enhance the transparency and completeness of the statistical analysis, it is suggested to provide more details regarding the selection of variables in the multivariate models and to discuss potential sources of bias. Additionally, a more detailed description of the health outcome prioritization tool used in the study, including its validity, reliability, and development, would be beneficial for readers to better evaluate the tool's utility and applicability in other clinical and research settings.

 Response: Many thanks for your comments and time in reviewing our manuscript.

We have now included further details regarding the statistical analyses and potential biases, and have provided our responses and location of the updates in the manuscript based on your individual comments below.

We have added further detail on the psychometric property testing previously carried out by the original developers on the tool, in the introduction section of the manuscript:

Introduction- Page 6: "Psychometric property testing was previously carried out by Fried et al, who found the tool was understood well by participants, with demonstrable construct validity and variable results in test re-test validity testing"

.

Furthermore, a dedicated section discussing the limitations of the study would be important to provide a critical view of the results, including discussions on potential selection biases, limitations of the methodology used, and issues related to the generalization of the findings. A discussion of the clinical and policy implications of the study results could also be included to highlight the practical relevance of the study, providing recommendations for clinical practice, public health policies, and directions for future research.

 Response: We have further elaborated on the practical relevance and clinical applications of this study in the "discussion" and "recommendations for the future sections" and also further elaborated regarding the limitations of this study in the "strengths and limitations" sections, all additions highlighted through track changes.

2 - Has the statistical analysis been performed appropriately and rigorously?

After reviewing the statistical analysis conducted in the manuscript, I noticed that the analysis was conducted appropriately, but there are areas where additional details and rigor could improve its comprehensiveness.

The text employs statistical techniques such as chi-square statistics, logistic regression models, and the Wilcoxon test to examine associations, differences, and changes in health priorities among participants. However, to provide a more comprehensive view of the statistical analysis, I have some suggestions for the authors:

Provide more details on the multivariate analysis, such as information on variable selection, consideration of interactions between variables, and interpretation of results in terms of the impact of different variables on health priorities.

Discussion of Potential Bias: A discussion of potential sources o

---

## [Decision Letter · Decision Letter 1]

1 Dec 2024

Health outcome priorities of people with multiple long-term conditions using the outcome prioritisation tool in the UK: a survey study and feasibility assessment

PONE-D-24-10844R1

Dear Dr. Sathanapally,

We’re pleased to inform you that your manuscript has been judged scientifically suitable for publication and will be formally accepted for publication once it meets all outstanding technical requirements.

Kind regards,

Filipe Prazeres, MD, MSc, Ph.D.

Academic Editor

PLOS ONE

Additional Editor Comments (optional):

Reviewers' comments:

Reviewer's Responses to Questions

**Comments to the Author**

1. If the authors have adequately addressed your comments raised in a previous round of review and you feel that this manuscript is now acceptable for publication, you may indicate that here to bypass the “Comments to the Author” section, enter your conflict of interest statement in the “Confidential to Editor” section, and submit your "Accept" recommendation.

Reviewer #1: All comments have been addressed

Reviewer #4: All comments have been addressed

2. Is the manuscript technically sound, and do the data support the conclusions?

Reviewer #1: Yes

Reviewer #4: Yes

3. Has the statistical analysis been performed appropriately and rigorously? 

Reviewer #1: Yes

Reviewer #4: Yes

4. Have the authors made all data underlying the findings in their manuscript fully available?

Reviewer #1: Yes

Reviewer #4: Yes

5. Is the manuscript presented in an intelligible fashion and written in standard English?

Reviewer #1: Yes

Reviewer #4: Yes

6. Review Comments to the Author

Reviewer #1: I would like to thank the authors for their sincere responses.

The manuscript is improved and I have no further comments.

Thank you.

Reviewer #4: Dear Authors,

Thank you for submitting your revised manuscript for review. I have carefully evaluated the changes and improvements made in response to the feedback provided. I am pleased to note that the modifications have significantly enhanced the clarity, coherence, and overall quality of the manuscript.

The revised sections effectively address the concerns raised during the initial review. The introduction now provides a more comprehensive background, the methodology is more clearly articulated, and the results are presented with greater precision and detail. Additionally, the discussion has been expanded to more thoroughly interpret the findings and their implications.

I appreciate the effort and diligence put into revising the manuscript, and I believe that it now meets the high standards required for publication. Therefore, I am happy to recommend that the manuscript be accepted for publication in its current form.

Congratulations on your excellent work.

7. PLOS authors have the option to publish the peer review history of their article (what does this mean?). If published, this will include your full peer review and any attached files.

Reviewer #1: No

Reviewer #4: No

---

## [Editor Report · Acceptance letter]

6 Dec 2024

PONE-D-24-10844R1 

PLOS ONE

Dear Dr. Sathanapally, 

I'm pleased to inform you that your manuscript has been deemed suitable for publication in PLOS ONE. Congratulations! Your manuscript is now being handed over to our production team.

Kind regards, 

on behalf of

Prof. Filipe Prazeres 

Academic Editor

PLOS ONE